# Apparent stability masks underlying change in a mule deer herd with unmanaged chronic wasting disease

Mark C. Fisher[1,3], Ryan A. Prioreschi[2,3], Lisa L. Wolfe[1], Jonathan P. Runge[1], Karen A. Griffin[1], Heather M. Swanson[2] & Michael W. Miller ![ORCID] [1✉]

The contagious prion disease "chronic wasting disease" (CWD) infects mule deer (*Odocoileus hemionus*) and related species. Unchecked epidemics raise ecological, socioeconomic, and public health concerns. Prion infection shortens a deer's lifespan, and when prevalence (proportion of adults infected) becomes sufficiently high CWD can affect herd dynamics. Understanding population responses over time is key to forecasting long-term impacts. Here we describe unexpected stability in prevalence and abundance in a mule deer herd where CWD has been left unmanaged. High apparent prevalence (~30%) since at least 2005 likely drove observed changes in the proportion and age distribution of wild-type native prion protein (*PRNP*) gene homozygotes among deer sampled. Predation by mountain lions (*Puma concolor*) may be helping keep CWD in check. Despite stable appearances, prion disease nonetheless impairs adult survival and likely resilience in this deer herd, limiting its potential for growth despite refuge from hunter harvest and favorable habitat and winter conditions.

[1] Colorado Division of Parks and Wildlife, 4330 Laporte Avenue, Fort Collins, Colorado 80521-2153, USA. [2] City of Boulder Open Space and Mountain Parks, 66 South Cherryvale Road, Boulder, Colorado 80302, USA. [3] These authors contributed equally: Mark C. Fisher, Ryan A. Prioreschi. ✉email: mike.miller@state.co.us

The contagious prion disease "chronic wasting disease" (CWD) infects mule deer (*Odocoileus hemionus*) and at least six other species of the deer (cervid) family[1–3]. Chronic wasting disease has been recognized for only a few decades, yet has become widely regarded as one of the most worrisome diseases affecting wildlife in the Northern Hemisphere[2–4]. Unchecked epidemics raise ecological, socio-economic, and public health concerns[2–4]. Prion infection shortens the host's lifespan, applying novel selective pressure and affecting herd dynamics when prevalence (proportion of adults infected) becomes sufficiently high[5–9]. Understanding how host populations respond over time is key to forecasting CWD's long-term impacts.

The chronic wasting disease is the only prion disease known to occur in free-living species and one of only two prion diseases shown thus far to be contagious. Perceived changes in the geographic distribution of CWD since the late 1990s, first within the North American continent and later expanding to include the Korean peninsula[10] and Scandinavia[11], have broadened trepidation about what was once an obscure wildlife disease believed confined to a small, contiguous portion of the western United States in Colorado and Wyoming[12]. Beyond its geographic extent, the array of natural host species – thus far at least seven, spanning four genera – suggests the potential for even larger-scale occurrence[2,3,13]. Some combination of host, agent, and environmental factors seem to drive efficient transmission[2,3,13], leading to epidemics where one in every three or four adult animals become infected annually[5,8,14]. At such levels, CWD can affect host population growth and stability[5–8].

Despite their explosive reputation, CWD epidemics in natural systems are slow to develop, unfolding over several decades in the wild[12,15]. The insidious dynamics, combined with relatively recent emergence or detection and lack of sustained interest and resources, have limited the opportunities to observe long term trends. To this end, we revisited the Table Mesa mule deer herd in southwest Boulder, Colorado USA, where we had last studied CWD in 2005–2009[5]. We gathered contemporary data for comparison to those from more than a decade earlier, assessing interim trends in disease prevalence, herd demography, and relative abundance of native prion protein (*PRNP*) genotype variants. Our analyses revealed unexpected stability in CWD prevalence and deer abundance despite changes in the proportion and age distribution of wild-type *PRNP* gene homozygotes likely driven by disease, as well as evidence that predation by mountain lions (*Puma concolor*) may be helping keep CWD in check at Table Mesa.

## Results and discussion

**Apparent epidemic and demographic stability.** Annual CWD incidence (new cases×total number in herd$^{-1}$ × year$^{-1}$) can be approximated by infection prevalence (proportion infected) in mule deer herds where the disease has become well-established[14]. Apparent prevalence (number positive×number sampled$^{-1}$; reported with 95% binomial confidence limits [bCL]) among sampled adult deer had reached 34% (25%–45%; $n = 91$) overall by winter 2018–2019 at Table Mesa, suggesting about one in three adult deer were now becoming infected each year. Male mule deer typically sustain prevalence twice that seen among sympatric females[5,8,14]. That gap had closed at Table Mesa since 2005 (Table 1). Overall, apparent prevalence had not changed measurably from that observed 13 years earlier (29% [20%–40%], $n = 85$; two-tailed Fisher's exact test $P = 0.522$).

On the surface, deer abundance and demography also appeared surprisingly unchanged considering the apparent duration of sustained high CWD incidence. Mark-resight surveys done in December 2018 estimated total deer abundance at 269 (95% CL 227–321) individuals, including 139 females, 64 males (M:F ratio ~46:100), and 67 juveniles. Essentially the same number of individuals had been estimated from surveys done in December 2005 using the same counting routes and methods[5]. Mean (±95% confidence interval) estimated ages among deer captured in 2018–2019 were 4.4 (±0.5) years for females and 3.5 (±0.3) years for males. Despite absence of sport hunting in the study area, none of the 46 males we captured exceeded 5.5 years of age. As with overall abundance, mean ages were similar to those estimated from deer captured in 2005 (females: 3.9 [±0.5] years; males: 4.1 [±0.6] years). This superficial stability masked more dynamic underlying changes within the herd.

**Prion-related genetic patterns.** In contrast to overall abundance and apparent CWD prevalence, the herd's observed genetic composition—as related to prion infection—had changed markedly since 2005. A polymorphism at codon 225 of the *PRNP* gene modulates prion disease course and relative infection risk in mule deer[8,13,16–19]. Genotype annotation reflects the deduced presence of serine (S) and/or phenylalanine (F) at codon 225[13,16]. Homozygous encoding for S at codon 225 (225SS) is considered the "wild-type" mule deer *PRNP* genotype[13,16]. This genotype shows higher relative infection probability and more rapid disease progression, although neither heterozygous (225SF) nor F-homozygous (225FF) deer are completely protected from infection despite lower infection risk[8,16–19]. Indeed, parsing by *PRNP* genotype (Table 1) revealed that prevalence among 225SS deer (~45%) was higher than among 225SF individuals (~7%; two-tailed Fisher's exact test $P = 0.007$).

Individuals of the 225SS genotype comprised a smaller overall proportion of the cross-sectional sample in 2018–2019 (~0.75, $n = 99$) than in 2005 (~0.92, $n = 92$; two-tailed Fisher's exact test $P = 0.0016$; Table 1, Fig. 1). The proportion of 225SF individuals in the cross-sectional sample had correspondingly increased, from 0.08 to 0.24. (One of the 99 deer genotyped in 2018–2019 was 225FF.) The new, lower frequency of 225SS individuals approximated that in Wyoming's South Converse mule deer herd, another heavily infected herd (apparent prevalence = 24%; proportion 225SS = 0.78)[8]. Moreover, the change in the observed frequency of 225*F (denoting SF or FF) genotypes at Table Mesa (from 0.08 to 0.24, or +0.16 over ~13 years) fell within the range predicted from a relationship between CWD prevalence and subsequent change in 225*F frequency observed 11–18 years later among mule deer in the South Converse and six other herds in Wyoming[19].

**Attrition vs. selection as the driver of apparent genetic trends.** Higher incidence and disease-associated mortality among the 225SS subpopulation likely drove the genetic patterns we observed. On first impression, the data create the illusion of a genetic shift when viewed overall. However, upon further scrutiny, we believe that the superficial pattern is better explained by earlier attrition of 225SS individuals in 2018–19 compared to the prior study period rather than by increased recruitment of 225F* = (SF or FF) individuals.

The cumulative effects of prion infection across age cohorts were evident. Although the proportion of 225SS deer recruited into the herd had remained high (~89%; 95% bCL 72–98%) since 2005, fewer 225SS deer persisted among the cohorts of deer aged 3 years or older sampled in 2018–2019 (Fig. 1). Moreover, proportionally fewer 225SS individuals 5 years of age or older were present in the 2018–2019 sample as compared to 2005 (Fig. 2). These findings lend support to our suggestion that attrition of 225SS individuals rather than greater recruitment of

**Table 1 Apparent prevalence of prion infection by *PRNP* genotype and sex in Table Mesa mule deer.**

| | 225SS | | | | 225SF | | | |
| | Female | | Male | | Female | | Male | |
| Year | n | Prevalence (bCL) | n | Prevalence (bCL) | n | Prevalence (bCL) | n | Prevalence (bCL) |
| --- | --- | --- | --- | --- | --- | --- | --- | --- |
| 2018–2019 | 40 | 38% (23–54%) | 31 | 48% (30–67%) | 7 | 14% (<1–58%) | 11 | 0% (0–29%) |
| 2005 | 55 | 24% (13–37%) | 25 | 44% (24–65%) | 4 | 0% (<1–60%) | 1 | 100% (3–100%) |

Apparent prevalence (proportion of sample infected) of chronic wasting disease infection among adult mule deer (*Odocoileus hemionus*) from the Table Mesa herd near Boulder, Colorado, USA. Results apportioned by *PRNP* codon 225 genotype (serine [S] or phenylalanine [F]) and sex. Prevalence in 2018–2019 did not differ between sexes, even among 225SS deer (two-tailed Fisher's exact tests P ≥ 0.468). Prevalence estimated from immunohistochemistry of rectal mucosa biopsies including ≥3 lymphoid follicles and reported with 95% binomial confidence limits (bCL). Data from 2005 are included for comparison[5]. Of the 100 deer sampled in 2018–2019, nine with inadequate biopsies, one 225FF, and one of an undetermined genotype are excluded from this tabular summary.

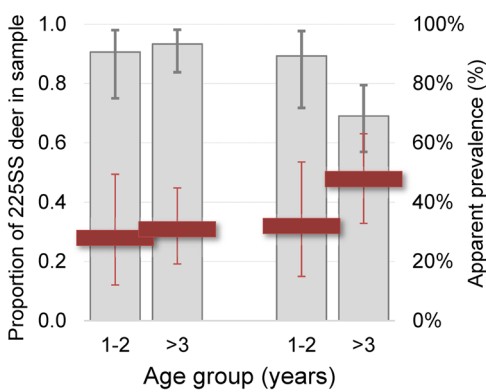

**Fig. 1 Change in the proportion of "wild type" *PRNP* homozygotes among older age Table Mesa mule deer.** The proportion of Table Mesa mule deer (*Odocoileus hemionus*) homozygous for serine (S) at *PRNP* codon 225 (225SS), or "wild type", in 1–2-year-old vs. older (≥3 years old) age cohorts sampled in 2005 or in 2018–2019 (grey bars) and corresponding prion infection prevalence (per cent of sampled deer testing positive) among 225SS deer in those cohort groups (red dashes). Individuals expressing at least one phenylalanine at codon 225 comprised the balance of deer sampled within each cohort; such individuals tend to have lower probability of prion infection and show slower disease progression[8,13,16–19]. The high proportion of 225SS deer (~0.89) among the cohort of 1- or 2-year-old deer (n = 28) indicate that their recruitment into the herd had not changed measurably between the two sampling periods. However, by 2018–2019 prion infection and subsequent disease-associated attrition of the relatively susceptible infected 225SS individuals ~2 years later manifested as a lower proportion of 225SS deer among the cohort of deer aged 3 years or older (n = 71) as compared to the younger cohort (two-tailed Fisher's exact test P = 0.042). Genetic data from deer sampled at Table Mesa in 2005[5] (n = 32 in 1–2-year-old cohort, n = 60 in ≥3-year-old cohort) are included for comparison to illustrate the relatively recent emergence of this pattern. Capped vertical lines are 95% binomial confidence intervals.

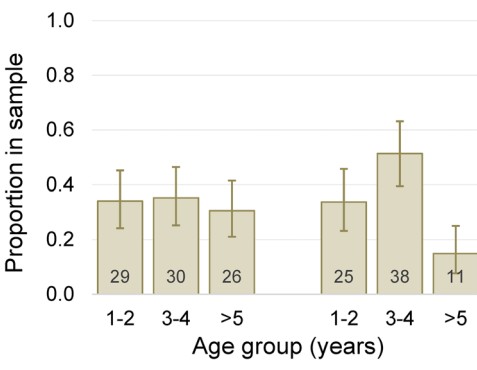

**Fig. 2 Change in age distribution among "wild type" *PRNP* homozygote mule deer at Table Mesa.** Proportional distribution of sampled Table Mesa mule deer (*Odocoileus hemionus*) homozygous for serine (S) at *PRNP* codon 225 (225SS)—"wild type"—across three age groups (1–2-year-old, 3–4 year old, or ≥5-year-old) in 2005 (n = 85) or in 2018–2019 (n = 74). The change in age distributions between the two study periods (2 × 3 Fisher's exact test P = 0.035) appeared to be driven by capture of fewer individuals ≥5 years old and an accompanying increase in 3–4-year-old individuals in the 2018–2019 sample. Because captures tended to focus on obviously adult animals our samples may have underrepresented the younger age class by largely excluding 1-year-olds, but this potential bias was consistent between the two study periods and would not give rise to differential capture/sampling probabilities between the two older age groups. Bars are the proportion of the sample in an age group from the respective study periods. Capped vertical lines are 95% binomial confidence intervals; numbers shown at the bottom of bars are respective sample sizes (number of individual deer).

225*F individuals may be responsible—at least in the short term—for observed genetic "shifts" in this and perhaps other CWD-infected mule deer herds. Marked changes in relative *PRNP* genotype abundance in our cross-sectional sampling occurred at Table Mesa within little more than a decade, approximating the timeframe for comparable changes reported in seven Wyoming mule deer herds[19]. Our findings may motivate consideration of demographic data in assessing and interpreting genetic patterns in other prion-infected cervid herds.

Although we cannot completely discount the possibility of another explanation, this pattern of attrition is consistent with observations of declining average age at infection and disease-associated death in the course of unchecked epidemics of CWD in captive mule deer[20] and scrapie in domestic sheep[21]. Considering about one of every three adult female deer at Table Mesa is

infected, the greater opportunity for exposure of younger females and males while still living in matriarchal social groups seems a plausible driver for the pattern observed. Relatively greater exposure within such groups also could be contributing to the diminished differences in apparent prevalence between the sexes in 2018–2019 (Table 1).

Prospects for an evolutionary rescue of this prion-infected mule deer herd seem dampened by the lack of an appreciable increase in 225*F genotype frequencies among younger age classes at Table Mesa despite marked 225SS attrition. Whether 225SS deer somehow maintained their reproductive success in the face of CWD attrition[22] or 225*F deer had lower recruitment cannot be discerned from our field data. However, *PRNP* polymorphisms likely have fitness-associated influences beyond modulating prion infection risk and disease progression[9,16,18,19,23]. In captivity, F-homozygous (225FF) mule deer appear to have inherently poor thrift and survival[18]. We expected to encounter more 225FF individuals in the contemporary Table Mesa herd given their longevity in the face of intense prion exposure[18], but such individuals remained rare.

Only one 225FF individual has been encountered at Table Mesa since 2005 (n = 231 total sampled). Similar scarcity occurred in the South Converse (2 of 143)[8] and other infected Wyoming mule deer herds (1 of 1,156)[19]. These field observations may lend support to the notion[18] that 225FF individuals have fitness disadvantages compared to wild type mule deer outweighing the benefits of prion disease resistance. The precise mechanism for a potential F allele effect on mule deer fitness is unknown. However, we noticed that ~36% (9/25) of the 225*F deer sampled had low (<5; ref. [24]) lymphoid follicle counts in their tissue biopsies, whereas only ~16% (12/74) of 225SS deer samples had low follicle counts (two-tailed Fisher's exact test P = 0.049). A PRNP polymorphism more broadly influencing the immune system could be an avenue for balancing selection[9,19] that may warrant further consideration and study.

**Lions and prions and deer.** As expected, prion infection lowered deer survival just as it had in our earlier study[5]. All 10 CWD-positive deer monitored via telemetry during 2018–2020 died within 415 days after capture. In contrast, only seven (~17%) of the 41 test-negative deer with telemetry died during that same timeframe. Mountain lion predation accounted for at least ten of the 21 mortalities among all 51 telemetry-monitored deer during 2018–2020. With annual mortality attributed to mountain lion predation running ~0.36 among infected deer vs. ~0.08 among apparently uninfected deer, infected deer showed relatively high vulnerability to predation similar to that observed previously at Table Mesa[5]. Only three of the 10 infected deer clearly succumbed to clinical CWD. Beyond the losses to predation and disease, vehicle collisions accounted for three deaths. For five, the cause of death could not be determined with certainty.

Mountain lions have been shown to selectively prey upon CWD-infected mule deer[25], and infected deer appear relatively vulnerable to predation[5,8]. Yet high CWD prevalence has persisted at Table Mesa for over a decade despite substantial mountain lion predation. On the surface, this observation may seem to undermine the notion that selective removal can facilitate disease suppression. However, such effects depend on the amount and timing of predatory removal relative to pathogen transmission and death resulting from infection itself[26–28]. Mountain lions killed at least 18 of the 53 CWD-infected 225SS deer monitored during our two field studies at Table Mesa. The survival distribution of 26 infected deer dying in end-stage CWD differed from that of infected deer killed by mountain lions (Mantel's logrank test, two-tailed, without continuity correction, P = 0.037; Fig. 3). Median survival was 215 days (95% CL 133–296 days) after capture among infected deer killed by mountain lions versus 343 days (95% CL 178–508 days) for those succumbing to clinical disease (Kaplan–Meier life table analysis).

The observed pattern seems consistent with the modest selective pressure expected from an ambush predator[25,27]. Early onset of prion shedding by infected mule deer[29,30] likely affords ample time for shedding to occur before infected individuals become vulnerable to predation. This outcome aligns with other field data and with modelling illustrating the extent of selective pressure needed to lower CWD incidence[27,31]. Human intolerance imposes bounds on mountain lion abundance[32,33] that may limit the potential impacts of predation on CWD incidence at Table Mesa and elsewhere. When of sufficient magnitude, sport hunting could complement natural predation by removing a larger proportion of the infected subpopulation[27,34]—its absence from the Table Mesa mule deer herd's range thus may constrain the overall expectations for lowering apparent prevalence and incidence. Nonetheless, predation by mountain lions seems likely to have influenced epidemic dynamics at Table Mesa to some

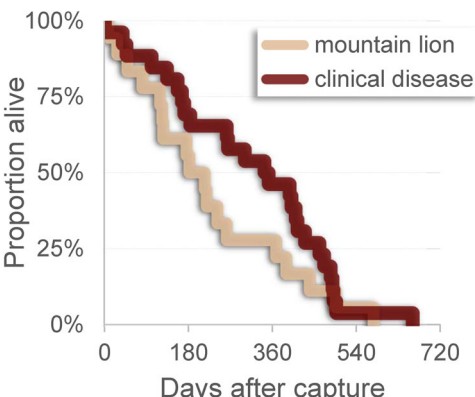

**Fig. 3 Survival curves for prion-infected mule deer ultimately killed by mountain lions or end-stage chronic wasting disease at Table Mesa.** Kaplin-Meier survival curves for chronic wasting disease (CWD)-infected mule deer (*Odocoileus hemionus*) killed by mountain lions (*Puma concolor*; lighter tan line) or succumbing to end-stage clinical disease (darker red line). Survival distributions for infected deer killed by mountain lions (n = 18) or clinical CWD (n = 26) differed (Mantel's logrank test, two-tailed, without continuity correction, P = 0.037). Data include infected deer homozygous for serine at PRNP codon 225 monitored at Table Mesa herd near Boulder, Colorado, USA, during 2005–2008 or 2018–2020.

extent and could have helped stabilize apparent prevalence since 2005.

## Conclusions

Reassessing the Table Mesa herd after more than a decade revealed unexpected superficial stability in apparent CWD prevalence and deer abundance in the time since our original study ended. Some combination of predation by mountain lions and perhaps subtle genetic shifting in the mule deer host or unidentified environmental factors may have contributed to the net absence of measurable change, at least on the surface. Despite the appearance of stability, the Table Mesa deer herd seems far from healthy. High disease incidence appears to be truncating the age distribution of the otherwise dominant wild-type individuals. Moreover, low adult survival sustained for over a decade likely impairs the resilience of this herd and limits its potential for growth despite an abundance of available habitat and relatively mild winter conditions. Periodic reassessment of this and other infected cervid herds will inform on long-term implications of CWD outbreaks to help better frame the policy choices surrounding intervention alternatives.

## Methods

**Deer capture and sampling.** We captured 100 mule deer (54 females, 46 males) during November 2018–February 2019, avoiding capture and sampling of juveniles. We attempted to distribute captures throughout the ~23 km² study area described by Miller et al.[5] to minimize spatial disparities in comparing contemporary and past data, and to assure marks were widely distributed for December ground counts to estimate deer abundance[5,35,36]. Field and sampling methods generally followed those used elsewhere[5,31,37]. Field procedures were reviewed and approved by the CPW Animal Care and Use Committee (file 14–2018).

We pursued deer on foot and darted them opportunistically, delivering sedative combinations intramuscularly via projectile syringe. Premixed immobilization drug combinations included either nalbuphine (N; 0.9 mg/kg) or butorphanol (B; 0.5 mg/kg) combined with azaperone (A; 0.2 mg/kg) and medetomidine (M; 0.2 mg/kg)[38], with standard total doses for respective combinations based on an estimated mass of 70 kg (average drug volume per animal was 1.3 ml NMA, 1.4 ml BAM). We collected rectal mucosa biopsies to determine CWD infection status[37]. We also collected whole blood and marked all deer with individually identifiable ear tags and some with telemetry (n = 51) or visual identification (n = 12) collars. Ages were estimated to the nearest year via tooth replacement and wear patterns[39]; observers used a pocket reference guide in the field to help assure consistency. To

antagonize sedation upon completion of handling and sampling, each deer received 5 mg atipamezole/mg M administered, injected intramuscularly.

**Prion diagnostics.** Formalin-fixed tissue biopsies were processed and analyzed by immunohistochemistry (IHC) at the Colorado State University Veterinary Diagnostic Laboratory (Fort Collins, Colorado USA; CSUVDL) for evidence of CWD-associated prion (PrP$^{CWD}$) accumulations using monoclonal antibody F99/97.6.1 (VMRD Inc., Pullman, Washington, USA)[40] and standard IHC methods[24,37,41], except that the CSUVDL's IHC staining machine (Leica Microsystems Inc., Buffalo Grove, Illinois, USA) was different from that used in earlier studies (Ventana Medical Systems, Oro Valley, Arizona, USA). Biopsies were evaluated microscopically and classified as positive (infected) or not detected (negative) based on PrP$^{CWD}$ presence or absence; the same pathologist (T. R. Spraker) read biopsies for both the current and prior[5] studies.

We included only data from deer with biopsies providing ≥3 lymphoid follicles in analyses involving infection status in order to maintain a relatively high (≥90%) probability of detecting infected individuals[24]. Two animals with low follicle counts that died shortly after capture were excepted by substituting postmortem IHC results. Limiting the acceptable follicle count excluded seven females (two 225SS, five 225SF) and two males (one 225SS, one 225SF) from some analyses. One male deer was 225FF and one female deer was missing a blood sample and thus not assigned to a *PRNP* gene group; these two individuals also were excluded from some analyses (e.g., Table 1).

**_PRNP_ genotyping.** We used DNA extracted from whole blood buffy coat aliquots (*n* = 99) to screen for the presence of sequences at *PRNP* gene codon 225 that encode for serine (S) and/or phenylalanine (F) in the mature prion polypeptide, classifying individuals as 225SS, 225SF, or 225FF[16,36,42]. Methods generally followed those described by Jewell et al.[16]. Briefly, we extracted DNA using the DNeasy® blood and tissue kit (Qiagen, Valenica, California, USA). We amplified the complete open reading frame (ORF) plus 25 bp of 5′ flanking sequences and 53 bp of 3′ flanking sequences in the *PRNP* coding region using polymerase chain reaction (PCR). Purified DNA was combined in a 0.2 ml PCR tube containing a puReTag Ready-To-Go PCR bead (illustra™, GE Healthcare Bio-Sciences Corp, Piscataway, New Jersey, USA). Each PCR bead contained 2.5 units puReTag DNA polymerase, 10 mM Tris-HCl, 50 mM KCl, 1.5 mM MgCl$_2$, 200 μM of each deoxynucleoside triphosphate, and stabilizers, including bovine serum albumin. For each PCR assay, 1 μL of each primer at 200 nM, 22 μL of RNase-free water and 1 μL of approximately 100 ng total genomic DNA was added for a final volume of 25 μL. Primers used for amplification were forward (MD582F, 5′-ACATGGGCA TATGTGCTGACACC-3′) and reverse (MD1479RC, 5′-ACTACAGGGCTGCA GGTAGATACT-3′) described by Jewell et al.[16]. Reactions were thermal-cycled in a PTC 100 (MJ Research) at 94 C for 5 min and then 32 cycles of 94 C for 7 s, 62 C for 15 s, 72 C for 30 s and a final cycle of 72 C for 5 min, and kept at 4 C until inspected for successful amplification by agarose gel electrophoresis. As confirmed by LaCava et al.[19], the MD582F and MD1479RC primers developed by Jewell et al.[16] specifically amplify the functional *PRNP* gene ORF, thereby excluding confounding effects that could arise from the presence of a processed pseudogene that occurs in a majority of deer (*Odocoileus* spp.)[42].

We used *Eco*RI restriction digestion of the PCR-amplified *PRNP* region[16]—a validated assay targeting the singular polymorphism at codon 225 in mule deer—to screen all 99 samples for presence of S or F codons. Aliquots (10 μl) of completed PCR reactions were incubated with 10 U *Eco*RI (New England Biolabs) in a total volume of 12 μl containing 50 mM NaCl, 100 mM Tris/HCl, 10 mM MgCl$_2$, 0.025% Triton X-100 (pH 7.5) at 37 C for 2–16 h followed by the addition of 2.5 μl 6× concentrate gel loading solution (Sigma- Aldrich) per sample, and the inspection of products by agarose gel electrophoresis for one 897bp-sized band for 225SS, two bands—one 897 bp and one 719 bp—for 225SF, or one 719 bp-sized band for 225FF. As noted by Jewell et al.[16], occurrence of TTC (the F codon) at position 225 creates an *Eco*RI recognition DNA sequence and cleavage site GAATTC from codons 224–225, whereas TCC (the S codon) creates the sequence GAATCC, which is not cut by *Eco*RI. When incubated with *Eco*RI, PCR products with a TTC codon at position 225 yielded cleavage fragments of the predictable sizes listed[16]. Because no other sites within the *PRNP* ORF DNA sequence are potentially transformable to GAATTC with one base change, this represents a specific genotyping method for assessing the S225F polymorphism in mule deer[16].

To confirm findings from *Eco*RI screening, we examined sequences of the complete *PRNP* ORF from 20 samples that showed evidence of cleavage indicating 225*F and 6 samples without cleavage identified as 225SS. For DNA sequencing, we used primers 245 (5′-GGTGGTGACTGACTGTGTGTTGCTTGA-3′), 12 (5′-T GGTGGTGACTGTGTGTTGCTTGA-3′) and 3FL1 (5′-GATTAAGAAGATAATG AAAACAGGAAGG-3′; Integrated DNA Technologies). Sanger sequencing was done on purified PCR product by Eurofins Genomics (Louisville, Kentucky, USA). Sequence chromatograms were viewed and DNA sequence alignments and comparisons were made using the MAFFT multiple sequence alignment program v7.450 module, software platform v2020.2.3 of Geneious Prime. Sequencing confirmed the presence of coding for F in all samples identified as 225*F by *Eco*RI digestion, as well as the absence of such coding in samples identified by *Eco*RI digestion as 225SS. Moreover, presence of AGC at codon 138 in all sequenced

samples reconfirmed that the primers we used had amplified the functional *PRNP* gene[42].

**Statistics and reproducibility.** For analyses, we tabulated IHC-positive and -negative results to estimate apparent prevalence of prion infection. We also tabulated the number of individuals assigned to *PRNP* genotypes and to age groupings as described. Age groupings were selected based on relevance to CWD epidemiology in mule deer[1,5,8,12,16–18,20,24,31,37]. Assuming a ~2-year disease course[5,8,17] and relative scarcity of end-stage disease in 225SS deer <2 years old[1,5,8,20], binning deer into 1–2-, 3–4-, and ≥5-year-old groups helped establish roughly when exposure and disease-associated losses were occurring. Binning also afforded a way to isolate the 1–2-year-old age group least likely to be affected by prion infection in other comparisons to earlier data. We compared proportions related to specific hypotheses using two-tailed Fisher's exact tests[43]. Descriptive statistics for survival data were derived from Kaplan–Meier life table analysis[43], and we compared survival curves for infected deer killed by mountain lions or succumbing to clinical disease using Mantel's logrank test (two-tailed, without continuity correction)[43]. For all comparisons, we used $\alpha = 0.05$ to ascribe differences.

Our sample of 54 female and 46 male deer approximated an a priori goal to compare data from 50 individuals of each sex based on original hypotheses about expected changes in apparent prevalence. (We expected prevalence to have increased, especially among females.) Sample sizes were limited but representative because the Table Mesa study herd is relatively small. Based on mark-resight results, we sampled nearly half of the non-juvenile individuals estimated to be present in the herd during winter 2018–2019. Large proportions of both sexes and younger adult age classes were represented even though the resulting numbers were somewhat small for estimating proportions with precision, especially when data were parsed across *PRNP* genotypes or age groupings.

We used data from biopsy sampling and mark-resight inventories done at Table Mesa in 2005 in assessing changes in apparent prevalence, *PRNP* genotype distributions, and deer abundance over time. In our earlier study[5], captures occurred during September–December and the infection status of deer was based on tonsil biopsy IHC. Because PrP$^{CWD}$ can be detected by IHC in tonsillar lymphoid follicles somewhat earlier in the disease course than in follicles associated with the rectal mucosa[37], our contemporary estimates of prevalence may be slightly underestimated. Despite differences in biopsy site, median survival of infected 225SS deer after capture did not differ between the two study periods (for 2005–08: 279 [95% CL 177–381] days, range 1–661 days, *n* = 44; for 2018–19: 215 [95% CL 95–335] days, range 27–415 days, *n* = 9; Mantel's logrank test, two-tailed, without continuity correction, $P = 0.173$).

Of the measurements taken, age estimation was perhaps the most vulnerable to potential bias. The vast majority of age estimates (92%) were made by one of four observers, including one (LLW) from the 2005 study. The use of pocket guides appeared helpful in assuring consistency among observers. Pairwise comparisons of the frequency distribution of age group assignments among the four main observers revealed no differences either overall ($\chi^2 \le 6.632$, Šidák-adjusted $P \le 0.199$) or relative to the most experienced observer (likelihood-ratio $\chi^2 \le 6.185$, $P \le 0.136$, with P values Bonferroni-adjusted for three comparisons).

**Reporting summary.** Further information on research design is available in the Nature Research Reporting Summary linked to this article.

## Data availability
Mule deer sex, age, *PRNP* genotype, biopsy results, and count data, as well as Sanger sequences used in genotype assay confirmation, are available from the Dryad Digital Data Repository: https://doi.org/10.5061/dryad.fbg79cnw6[44].

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

## Acknowledgements

Our work was supported by the Colorado Division of Parks and Wildlife (CPW), the City of Boulder, and US Fish and Wildlife Service (Pittman-Robertson) Federal Aid in Wildlife Restoration funding. We thank W. Keeley, C. Nunes, C. Fairbanks, M. Yarbrough, S. Gray, D. Gustafson, E. Collins, K. Cannon, P. Boyatt, and multiple other agency staff members for field assistance with deer capture, inventory, and monitoring, and T. Spraker and others at the Colorado State University Veterinary Diagnostic Laboratory for prion IHC, and K. Fox for assistance with PRNP gene sequencing. We also appreciate the critiques, comments, and suggestions offered to improve earlier versions. Finally, we extend sympathy to the Table Mesa neighbourhood, greater Boulder community, and especially victims' families in the wake of the tragedy suffered there on 22 March 2021.

## Author contributions

M.W.M., L.L.W., and H.M.S. conceived this study. All co-authors participated in study planning and in one or more aspects of fieldwork or laboratory analyses. M.C.F. and R.A.P. coordinated fieldwork and managed data. L.L.W., H.M.S., and M.W.M. provided prior data and ensured continuity between this and the earlier study at Table Mesa. K.A.G. conducted and verified PRNP genotyping. J.P.R. conducted or reviewed data analyses. M.W.M. and L.L.W. wrote the first draft of the manuscript and H.M.S., M.C.F., R.A.P., J.P.R., and K.A.G. reviewed and offered edits to shape subsequent versions.

## Competing interests

The authors declare no competing interests.
