## [Peer Review File · Communications Biology]

Reviewers' comments:

Reviewer #1 (Remarks to the Author):

This is a fairly straight-forward study revisiting a mule deer herd studied a decade previously for chronic wasting disease. Here, they examine potential changes for this population in CWD prevalence, herd demography, and genotype frequencies for the prion protein. They find that prevalence and demography are relatively unchanged over the last decade but genotype frequencies have changed. This study is valuable in that it helps evaluate long-term trends and impacts of this important wildlife disease.

The genetic piece is particularly interesting, but I feel I'm missing part of the story. They emphasize the overall proportion of S homozygotes has gone down in the last decade as CWD prevalence has remained high. But according to Fig 1, this seems to be true only for those individuals 3 years old and older. The proportion of 225SS among 1 to 2 year-olds remained unchanged and high at 90%. This is surprising given that the older animals that contribute more to reproduction have a higher proportion of individuals with the F allele. It would be interesting to calculate how many you would expect at Hardy-Weinberg equilibrium based on the breeding population. One of the shortcomings is not having genetic samples from juveniles, so we could tell whether there is lower juvenile survival among 225*F individuals or lower reproduction of adults. This all assumes that there really is a high proportion of SS among 1 to 2 year-olds. It would be helpful to show the sample sizes or confidence intervals of the proportion homozygous of the different age classes so we could judge confidence in this metric.

They say that only biopsies with >5 lymphoid follicles were included in the analyses, but they also say that the 225*F deer had low lymphoid follicle counts. This could bias the prevalence estimates among those with the F allele. This needs to be addressed.

Based on their results, mountain lion predation may reduce the infectious period by 130 days or so on average. In Line 124 they say that mountain lion predation doesn't appear to measurably suppress transmission, but I don't they can say that. Prevalence may very well be lower than it would be without lion predation. It would be interesting to model the expected R0 and equilibrium prevalence with and without this amount of mountain lion predation (with a 130 day reduction in infectious period), though I don't expect it as part of this study.

They talk about selective pressure of predation here, but that's not what they show. They do not show us the difference between predation rates on infected vs uninfected. (Perhaps this was in the previous study, but it's not mentioned here.)

They suggest in the abstract that a combination of changing gene frequencies and selective predation may have led to unchanged disease prevalence or deer abundance. This doesn't make sense to me. With chronic diseases, the system eventually reaches an equilibrium prevalence, and perhaps this system has reached equilibrium. They didn't show any results on the predation side to suggest that there have been changes in predation over the last decade, which could change the equilibrium. Then the question is why haven't changes in gene frequencies that seem to affect susceptibility not led to significant changes in prevalence?

Line 41: They define CWD incidence here as new cases/ year, but in the publication they reference, the definition was new cases/ population size/ year?

In summary, there are some intriguing results here, but not enough information to fully understand why these patterns have arisen. Nevertheless, it is a valuable contribution.

Reviewer #2 (Remarks to the Author):

The manuscript by Fisher et al. provides valuable insight into the longitudinal impact of CWD on mule deer populations. The methods implemented are standard and the paper advances our knowledge of the population genetics surrounding PRNP variation within a CWD endemic zone.

Access to such longitudinal samples is extremely rare and therefore the work has a high degree of novelty. I recommend the manuscript ultimately be published. However, I have comments that should be addressed before further consideration for publication:

1) The authors utilized IHC analyses of rectal biopsies to confirm CWD status. Can the authors elaborate on the sensitivity and specificity of rectal biopsies, especially pertaining to PRNP variation of mule deer? Previous work (Thomsen et al. 2012: Diagnostic accuracy of rectal.....; although in white-tailed deer) indicates rectal biopsy-based IHC diagnostic sensitivity and specificity is directly influenced by genotype. What is the potential for false negatives / false positives using this testing regime? Were the same CWD testing methods / antibodies used between the 2005 study and the 2018/2019 study? Please clarify within the methods section.

2) Although the authors cite the Brayton et al. 2004 manuscript reporting a mule deer PRNP pseudogene, there is no mention of how the team avoided potentially spurious results due to a pseudogene. The authors should specify their primer selection and how they avoided the pseudogene. Moreover, how were the PRNP genotypes confirmed? Clarification of these methods would be appreciated.

3) The authors should provide the IACUC (or equivalent animal care and use regulatory body) protocol under which the research was performed.

Reviewer #3 (Remarks to the Author):

The paper titled 'Lions and Prions and Deer Reprise' by Fisher et al. describes chronic wasting disease and host genotype dynamics in a wild mule deer population. CWD is now widely distributed in the US (25 States have reported CWD in their deer populations as of January 2021). As the geographic spread of CWD continues, management of CWD in wild and captive cervid populations is becoming increasingly important for wildlife agencies across North America. Studies like this one are especially critical as they provide valuable insights about the long-term dynamics of CWD and have the potential to inform meaningful CWD management strategies.

Here, I list some concerns and provide my suggestions:

1) Line 41: "Annual CWD incidence (new cases/year) can be approximated by infection prevalence in mule deer herds."

I think it is important to specify the epidemiological context for this statement. Such approximations are possible when CWD is established in a population and prevalence is high but will not work when CWD cases are clustered, and prevalence is low (before CWD is established in a population or region).

2) Line 42: "Prevalence (95% binomial CL) had reached 37% (27%–48%; n=81) overall by winter 2018–2019 at Table Mesa, suggesting more than 1 in 3 adult deer were now becoming infected each year." Here you have n= 81, but in Table 1 the total number of samples for 2018-2019 add to 79.

3) You have used the term prevalence throughout the manuscript - I am assuming you are reporting apparent prevalence here and have not factored in the sensitivity/specificity of the diagnostic method (immunohistochemistry of rectal mucosa biopsies, right?). The term apparent prevalence only appears in Table 1 (line 261). I think this should be explicitly stated early on in the manuscript.

4) Line 62: "Parsing by PRNP genotype revealed that prevalence among 225SS deer (~45%) was nearly 7× higher than among 225SF individuals (~7%; two-tailed Fisher's exact test P=0.007)." This comparison is somewhat questionable (7x higher) as the sample size for 225SF individuals is low (1 CWD+ in 15 tested, 7% apparent prevalence with 95%CI of 0.1% to 32%). Please consider rephrasing lines 63-65 : "Higher incidence and disease-associated mortality among the 225SS subpopulation likely drove the genetic shift we observed."

5) Line 72: "Serial data on prevalence and PRNP genotype frequency may be useful for estimating the time since disease introduction in newly discovered CWD foci." Please reassess this sentence, because for any 'newly discovered CWD foci' there will be a lack of serial data on prevalence as CWD prevalence is low (<1%) for 10-15 years post introduction and CWD is difficult to detect during this period. Consequently, serial data on prevalence will be generated several years after CWD discovery and CWD introduction. The first detection in Colorado wild mule deer was 1985, right?

6) Your discussion about the role of mountain lion predation on the pattern of CWD in deer populations is intriguing (lines 107-125), but please consider adding an important detail to this discussion - the Table Mesa herd is not harvested and is therefore unlike most wild mule deer populations in Colorado. Hunter harvest in other mule deer populations possibly influence CWD dynamics to a larger extent compared to mountain lion predation in this study population.

Unless otherwise noted, the full citations for papers cited in responses are included in the manuscript's References.

Referee #1 – disease ecology, large mammal pred-prey dynamics – Reviewer #1 (Remarks to the Author):	Authors' response
This is a fairly straight-forward study revisiting a mule deer herd studied a decade previously for chronic wasting disease. Here, they examine potential changes for this population in CWD prevalence, herd demography, and genotype frequencies for the prion protein. They find that prevalence and demography are relatively unchanged over the last decade but genotype frequencies have changed. This study is valuable in that it helps evaluate long-term trends and impacts of this important wildlife disease.	We appreciate this overall assessment.
The genetic piece is particularly interesting, but I feel I'm missing part of the story. They emphasize the overall proportion of S homozygotes has gone down in the last decade as CWD prevalence has remained high. But according to Fig 1, this seems to be true only for those individuals 3 years old and older. The proportion of 225SS among 1 to 2 year-olds remained unchanged and high at 90%. This is surprising given that the older animals that contribute more to reproduction have a higher proportion of individuals with the F allele. It would be interesting to calculate how many you would expect at Hardy-Weinberg equilibrium based on the breeding population. One of the shortcomings is not having genetic samples from juveniles, so we could tell whether there is lower juvenile survival among 225*F individuals or lower reproduction of adults. This all assumes that there really is a high proportion of SS among 1 to 2 year-olds. It would be helpful to show the sample sizes or confidence intervals of the proportion homozygous of the different age classes so we could judge confidence in this metric.	The genetic observations were surprising & did not surface until we were exploring potential explanations for the scarcity of 225FF individuals. In retrospect, having genetic data on juveniles likely would have been illuminating and we have considered doing follow-up work. We have modified the revised text for consistency. On first impression, the data do create the illusion of a genetic shift when viewed overall. However, after further scrutiny that superficial pattern seems best explained by more rapid attrition of 225SS individuals in 2018-19 compared to the earlier study period rather than by increased recruitment of 225F* (= SF or FF) individuals. Additional analyses done in the course of revision revealed differences in the age distributions among 225SS deer between 2005 & 2018 that support this explanation. The phenomenon of progressively earlier onset & mortality has been described in sustained epidemics of chronic wasting disease in captive mule deer and in scrapie-infected sheep flocks (references added to the revised paper). We have added sample sizes and confidence intervals as suggested. Figure 1 has been reconstructed & simplified to accommodate transparency & data interpretation. Although not included in the text, the Hardy-Weinberg equilibrium based on the current breeding population would be [0.757 SS : 0.226 SF : 0.017 FF]; the confidence interval surrounding the observed proportion of 225SS among 1-2 yr olds does cover the H-W estimate, but not by much... We added a new figure and discussion to show how (& perhaps why) the age structure within the 225SS subpopulation appears to have changed. We also have reassessed (& replaced in some instances) the use of the term "shift" throughout. We hope this addresses the points raised.

They say that only biopsies with >5 lymphoid follicles were included in the analyses, but they also say that the 225*F deer had low lymphoid follicle counts. This could bias the prevalence estimates among those with the F allele. This needs to be addressed.

We understand the basis for the concern raised. We try to avoid tossing hard-won field data, and did so in this case with reservations. Ironically, we excluded samples with low follicle counts in the original analyses in order to *minimize* bias. Early in the course of prion infection in mule deer, disease-associated prion protein (PrP^d) deposits are small & scattered in lymphoid tissues; moreover, deposits appear a bit later in the disease course in rectal mucosa lymphoid tissue as compared to other sites. Geremia et al. (2015; their Fig. 6 appears at right) showed that the probability of a false-negative test result increases in biopsy samples with low follicle counts. Thus including a large number of low-follicle samples (= potential false-negative individuals) could lead to underestimating true prevalence, especially in adult deer.

Our original decision to use the 5 follicle cut-off was made *a priori*. We were surprised (& disappointed!) to see so many low count samples. And we concur that this did disproportionately lower the sample size of 225SF individuals. After reviewing the Geremia et al. (2105) paper and considering all of the reviewers' comments, we believe lowering the cut-off to ≥ 3 follicles offers a means of including additional samples in the analyses without unduly biasing prevalence estimates. Having at least 3 follicles to examine should keep the probability of false-negative results below ~ 0.1 . We have included additional Methods text to explain this.

The choice of follicle cut-off value had relatively little impact on estimates (see table below; 95% binomial confidence limits shown). Making the adjustment to include results from all samples with ≥ 3 follicles did not alter any of our original conclusions.

follicle cut-off	225SF		225SS	
	n (pos)	prev (bCL)	n (pos)	prev (bCL)
≥ 1	24 (1)	0.04 (0.001-0.21)	74 (30)	0.41 (0.23-0.53)
≥ 3	18 (1)	0.06 (0.001-0.27)	71 (30)	0.42 (0.31-0.55)
≥ 5	15 (1)	0.07 (0.002-0.32)	64 (29)	0.45 (0.33-0.58)

Based on their results, mountain lion predation may reduce the infectious period by 130 days or so on average. In Line 124 they say that mountain lion predation doesn't appear to measurably suppress transmission, but I don't they can say that. Prevalence may very well be

The original statement was based on the lack of measurable decline in prevalence, but we concur that predation may have contributed to the relative stability observed. The revised text is more circumspect with respect to the potential effects of mountain lion predation on the observed prevalence pattern, instead noting that

lower than it would be without lion predation. It would be interesting to model the expected R0 and equilibrium prevalence with and without this amount of mountain lion predation (with a 130 day reduction in infectious period), though I don't expect it as part of this study.	although transmission was not completely extinguished, removals by mountain lions may have contributed to the observed stability. This concern also was expressed in a comment farther below.
They talk about selective pressure of predation here, but that's not what they show. They do not show us the difference between predation rates on infected vs uninfected. (Perhaps this was in the previous study, but it's not mentioned here.)	Correct, selective predation & relative vulnerability were described previously, as referenced in lines 107-8. Although not as robust given the smaller number of radio collared animals tracked during 2018-20, the annual loss to mountain lion predation was ~4.2× higher among infected deer (0.35 vs. 0.08 for uninfected deer) and essentially the same as the ~4× greater vulnerability observed in our earlier study. We removed the term "selective" in line 123. We also added (~line 108 in original text) brief mention that the observed pattern of differential vulnerability of infected deer to mountain lion predation is essentially the same (~4×) as observed in our earlier study to further address this comment.
They suggest in the abstract that a combination of changing gene frequencies and selective predation may have led to unchanged disease prevalence or deer abundance. This doesn't make sense to me. With chronic diseases, the system eventually reaches an equilibrium prevalence, and perhaps this system has reached equilibrium. They didn't show any results on the predation side to suggest that there have been changes in predation over the last decade, which could change the equilibrium. Then the question is why haven't changes in gene frequencies that seem to affect susceptibility not led to significant changes in prevalence?	Based on prior observations at Table Mesa and experiences to date with this disease, we were surprised that apparent prevalence had not changed. Equilibrium is not a given with CWD, and things can get much worse: in Saskatchewan, for example, "prevalence rates across the province have continued to increase, ranging up to 70 per cent in mule deer...". Although we cannot discount the possibility, the Table Mesa system is or has been in some form of equilibrium, it also seems possible that the epidemic has reached a phase where measures of overall prevalence and abundance no longer reflect underlying patterns as well as they do in early stages. Perhaps prevalence is not a good stand-alone metric for long-term tracking of CWD epidemics in the wild. In particular, trends among the 225SS subpopulation suggest the epidemic remains somewhat dynamic despite superficial appearances to the contrary. In short, status quo may not equate to true equilibrium. We believe the revised text and added details better describe & discuss the patterns we observed.
Line 41: They define CWD incidence here as new cases/ year, but in the publication they reference, the definition was new cases/ population size/ year?	Corrected in revised text as (new cases×total number in herd⁻¹×year⁻¹). Thanks for catching that.
In summary, there are some intriguing results here, but not enough information to fully understand why these patterns have arisen. Nevertheless, it is a valuable contribution.	Although the data presented are not sufficient to explain the patterns observed in full, we do believe the observations themselves offer insights that may help motivate hypotheses and studies better positioned to provide such explanations.

Referee #2 – genomic approaches to disease ecology, emergence of zoonoses, prion diseases – Reviewer #2 (Remarks to the Author):	
The manuscript by Fisher et al. provides valuable insight into the longitudinal impact of CWD on mule deer populations. The methods implemented are standard and the paper advances our knowledge of the population genetics surrounding PRNP variation within a CWD endemic zone. Access to such longitudinal samples is extremely rare and therefore the work has a high degree of novelty. I recommend the manuscript ultimately be published. However, I have comments that should be addressed before further consideration for publication:	We appreciate the overall assessment and questions raised.
1) The authors utilized IHC analyses of rectal biopsies to confirm CWD status. Can the authors elaborate on the sensitivity and specificity of rectal biopsies, especially pertaining to PRNP variation of mule deer? Previous work (Thomsen et al. 2012: Diagnostic accuracy of rectal.....; although in white-tailed deer) indicates rectal biopsy-based IHC diagnostic sensitivity and specificity is directly influenced by genotype. What is the potential for false negatives / false positives using this testing regime? Were the same CWD testing methods / antibodies used between the 2005 study and the 2018/2019 study? Please clarify within the methods section.	These are fair points. To clarify, we used biopsy results to assign infection status based on whether not abnormal prions deposits were detectable at the time of capture. A true immunohistochemistry (IHC)-positive biopsy does appear to be a reliable indicator of prion infection in mule deer. However, we did not regard negative biopsies as absolutely confirming an uninfected status. This is in keeping with the more general rule that neither rectal mucosa nor tonsil biopsies are used to “confirm” freedom from infection for reasons demonstrated by Thomsen et al. (2012) and others. Confirmatory testing is done using retropharyngeal lymph node tissue samples collected postmortem and tested via the same IHC assay, although even this approach has some limitations on detecting very early infections. We acknowledge the basis for the question & reworded some text to clarify this point (e.g., “CWD-negative” changed to “test-negative” where appropriate in context). The known PRNP polymorphisms in either mule deer or white-tailed deer can affect biopsy/IHC sensitivity but would not be expected to affect specificity. Thomsen et al. (2012) reported overall specificity >99%, and in general this immunohistochemistry assay is highly specific in deer. We are not aware of any potential for false positive results associated with one genotype or another. With respect to sensitivity, rectal mucosa lymphoid tissue becomes positive somewhat later in the disease course than more rostral lymphoid tissues. So biopsy would be somewhat less sensitive in detecting the earliest cases and our sampling approach likely did underestimate true prevalence to some extent. That could complicate comparisons to data based on postmortem sampling, but would have less effect on inferences made within the Table Mesa data set or comparisons to other

field studies reporting biopsy data. Delayed detection likely would be exacerbated in 225SF individuals because disease progression is slower (see Fox et al. 2006 for mule deer). The expected pattern seems likely to be similar to that reported by Thomsen et al. (2012; see their Fig. 6) for G96S polymorphism effects in white-tailed deer; also see Wolfe et al. (2007) for comparable serial antemortem sampling in white-tailed deer and in 225SS mule deer. Thus it's conceivable that prevalence among 225*F deer was >1.5× higher than we (& others) have reported, although this seems unlikely because the Thomson et al. (2012) data set was heavily biased toward sampling animals in early ("preclinical") disease stages & consequently accentuating genetic differences. But natural infection in 225*F mule deer is relatively rare based on postmortem sampling (e.g., Jewell et al. 2005, LaCava et al. 2021). Consequently, most "not detected" results in a cross-sectional sample would be correct, thereby lessening the consequences of potential biopsy IHC insensitivity. (See figure above from Geremia et al. 2015 for an illustration of this principle applied to younger vs. older deer.) The 225SS deer in our sample were ~13× more likely to test positive than 225*F deer (OR ~13.2, 95% exact CI 1.8–566.7). This approximates the odds reported by Jewell et al. (2005) for postmortem test data from individual mule deer herds, suggesting our biopsy data are not severely biased.

One way to maximize detection probabilities in antemortem testing (and thus maximize sensitivity to the extent possible) is to censor samples with relatively few lymphoid follicles available for examination. As noted above, we originally used 5 follicles as the cut-off but have dropped that to 3 follicles (~90% detection probability) for the revised analyses to address concerns about potential bias given the high proportion of 225SF deer with low counts.

According to the reference laboratory, the same IHC methods and monoclonal antibody (mAb) were used for screening in 2005 and 2018/19. The same pathologist (T. Spraker) read slides for both studies. We recently became aware that the staining machine used for IHC had changed (ca. 2012), but the new system is certified to meet federal (US) standards for prion diagnostics. In the course of reviewing methods, however, we noticed that IHC data from 2005 used tonsil biopsies. Wolfe et al. (2007) reported nominally lower detection from rectal biopsies (~88% vs. 100%; two-sided paired Fisher's exact test $P=0.5$; $n=17$) in 225SS deer orally inoculated 253 days earlier (dpi); results converged by ~130 days later. Using rectal biopsy in 2018/19 thus could have slightly diminished our ability to detect differences between periods.

We have clarified details & potential differences in the revised Methods text.

2) Although the authors cite the Brayton et al. 2004 manuscript reporting a mule deer PRNP pseudogene, there is no mention of how the team avoided potentially spurious results due to a pseudogene. The authors should specify their primer selection and how they avoided the pseudogene. Moreover, how were the PRNP genotypes confirmed? Clarification of these methods would be appreciated.	As noted by Brayton et al. (2004; Fig. 3), the processed PRNP pseudogene is monomorphic for serine (S) at codon 225 in mule deer. The EcoRI recognition and cleavage site used in screening for phenylalanine (F) only occurs in the functional gene (Brayton et al. 2004, Jewell et al. 2005), thus precluding false 225 F* signals arising from the pseudogene. More importantly, the primers we used – originally described by Jewell et al. (2005) – were specific to amplifying the functional PRNP gene. This was confirmed through correspondence with Dr. Jewell (retired), as well as with Drs. LaCava & Ernest (University of Wyoming) who confirmed >1,000 mule deer PRNP sequences from PCR products derived using these same primers. Per Dr. LaCava: We read about the pseudogene in Brayton et al. 2004, which states, "The pseudogene alleles invariably encoded N (aac) at codon 138". So we checked whether our PRNP sequences showed variation at codon 138, and all of our 1,000+ sequences had the reference sequence (AGC) and none had the pseudo-gene sequence (AAC). In addition, and also to address the question about confirmation of our genotyping results, we sequenced the PCR products from all 225*F deer and a subset of 10 225SS individuals. Sequencing confirmed the PCR products we analyzed here were all from the functional PRNP gene and that the EcoRI digest method reflected S225F status based on sequencing. We have added details to the Methods as suggested, including the primer sequences and confirmatory sequencing.
3) The authors should provide the IACUC (or equivalent animal care and use regulatory body) protocol under which the research was performed.	This information was included in Ln 242-3 of the original submission: "Field procedures were reviewed and approved by the CPW Animal Care and Use Committee (file 14-2018)." Per the formatting instructions, this information is now included in the Methods section of the revised manuscript.

Referee #3 (Aniruddha Belsare) – disease ecology – Reviewer #3 (Remarks to the Author):	
The paper titled 'Lions and Prions and Deer Reprise' by Fisher et al. describes chronic wasting disease and host genotype dynamics in a wild mule deer population. CWD is now widely distributed in the US (25 States have reported CWD in their deer populations as of January 2021). As the geographic spread of CWD continues, management of CWD in wild and captive cervid populations is becoming increasingly important for wildlife agencies across North America. Studies like this one are especially critical as they provide valuable insights about the long-term dynamics of CWD and have the potential to inform meaningful CWD management strategies. Here, I list some concerns and provide my suggestions:	We appreciate the assessment and suggestions for improvement.
1) Line 41: "Annual CWD incidence (new cases/year) can be approximated by infection prevalence in mule deer herds." I think it is important to specify the epidemiological context for this statement. Such approximations are possible when CWD is established in a population and prevalence is high but will not work when CWD cases are clustered, and prevalence is low (before CWD is established in a population or region).	This relationship holds across a wide range of apparent prevalence values, including values considered by some to be relatively "low". Nonetheless, point taken that disease was well established in all of the herds included in the assessment of this relationship. We have modified the text to qualify this relationship as holding "...in mule deer herds where the disease has become well-established."
2) Line 42: "Prevalence (95% binomial CL) had reached 37% (27%–48%; n=81) overall by winter 2018–2019 at Table Mesa, suggesting more than 1 in 3 adult deer were now becoming infected each year." Here you have n= 81, but in Table 1 the total number of samples for 2018-2019 add to 79.	The discrepancy is explained by the inclusion of two additional animals in the overall estimate that do not appear in Table 1. (One of the extra animals was 225FF, the other had missing genotype data because blood was not collected.) We have noted in the revised methods & table that data from these two animals also were included in some calculations. Apologies for the confusion.
3) You have used the term prevalence throughout the manuscript - I am assuming you are reporting apparent prevalence here and have not factored in the sensitivity/specificity of the diagnostic method (immunohistochemistry of rectal mucosa biopsies, right?). The term apparent prevalence only appears in Table 1 (line 261). I think this should be explicitly stated early on in the manuscript.	Agreed & corrected as suggested (e.g., in abstract [line11], line 41, etc.).

4) Line 62: "Parsing by PRNP genotype revealed that prevalence among 225SS deer (~45%) was nearly 7× higher than among 225SF individuals (~7%; two-tailed Fisher's exact test P=0.007)." This comparison is somewhat questionable (7x higher) as the sample size for 225SF individuals is low (1 CWD+ in 15 tested, 7% apparent prevalence with 95%CI of 0.1% to 32%). Please consider rephrasing lines 63-65 : "Higher incidence and disease-associated mortality among the 225SS subpopulation likely drove the genetic shift we observed."	Text revised after adding data as recommended by Reviewer #1 (above). We removed the multiplier in deference to the point about wide confidence limits surrounding the apparent prevalence among 225SF deer. As above, we also have better explained what we believe to be the drivers of the patterns observed & reconsidered use of the term "shift" throughout the paper.
5) Line 72: "Serial data on prevalence and PRNP genotype frequency may be useful for estimating the time since disease introduction in newly discovered CWD foci." Please reassess this sentence, because for any 'newly discovered CWD foci' there will be a lack of serial data on prevalence as CWD prevalence is low (<1%) for 10-15 years post introduction and CWD is difficult to detect during this period. Consequently, serial data on prevalence will be generated several years after CWD discovery and CWD introduction. The first detection in Colorado wild mule deer was 1985, right?	Point taken. We struck this sentence from the revised text.
6) Your discussion about the role of mountain lion predation on the pattern of CWD in deer populations is intriguing (lines 107-125), but please consider adding an important detail to this discussion - the Table Mesa herd is not harvested and is therefore unlike most wild mule deer populations in Colorado. Hunter harvest in other mule deer populations possibly influence CWD dynamics to a larger extent compared to mountain lion predation in this study population.	Agreed. We have added this point to the revised text and rearranged that paragraph to address both this and an earlier comment.

REVIEWERS' COMMENTS:

Reviewer #1 (Remarks to the Author):

Several of my previous comments have been addressed, but there are still a few issues which need to be clarified.

The most interesting part of this study was the genetic piece, but that story has somewhat changed since the last version. They have (I think rightly) removed the term 'genetic shift', because although they see an decrease in the overall proportion of 225SS, it doesn't actually appear to be a genetic shift due to natural selection, since the younger age classes still have high frequencies. They conclude that the decrease in SS individuals is due to greater attrition of SS due to higher incidence and mortality. (This was actually clearer in the rebuttal letter than in the main text, because in the text the idea is spread out over several paragraphs with other concepts in between.) But the interesting yet glossed over part is that it isn't translating to increased *F frequencies in younger age classes. We can't tell why – if 225SS are dying after they reproduce, or if recruitment is lower for *F. But there needs to be more of a discussion of this. It is actually a really important finding and suggests that we might not expect evolution to come to the rescue here.

Additionally, a new study is referenced here now (ref 19), which is also comparing past and current trends at a site in Wyoming. There needs to be more of a discussion of how these 2 studies fit together. For example, they saw comparable changes in genotype, but didn't they also see an increase in prevalence? They didn't look at age structure. Based on your findings of higher attrition of SS individuals rather than a genetic shift, could this change the implications of their study?

225SS are implied to have lower survival because we aren't seeing as many older individuals as before. If survival was lower in 2018-2020 compared to 2005-2008, they shouldn't be lumped together in Fig 3. They should be examined separately.

"The phenomenon of progressively earlier onset & mortality has been described in sustained epidemics of chronic wasting disease in captive mule deer and in scrapie-infected sheep flocks (references added to the revised paper)."

In these previous studies was prevalence increasing or remaining the same?

Line 152: This line is unclear. Is 0.16 the change in frequencies (A 16% increase?) Or did it change to 16% of these genotypes? If so where is the missing 14%? (Unidentifiable?)

Reviewer #2 (Remarks to the Author):

The authors have addressed my comments. I recommend the manuscript be accepted for publication.

Unless otherwise noted, the full citations for papers cited in responses are included in the manuscript's References.

In addition to the changes noted here, we made further edits in this final revision to comply with the instructions provided by the Associate Editor.

Referee #1 – disease ecology, large mammal pred-prey dynamics – Reviewer #1 (Remarks to the Author):	Authors' response
Several of my previous comments have been addressed, but there are still a few issues which need to be clarified.	We appreciate the additional suggestions for improvement.
The most interesting part of this study was the genetic piece, but that story has somewhat changed since the last version. They have (I think rightly) removed the term 'genetic shift', because although they see an decrease in the overall proportion of 225SS, it doesn't actually appear to be a genetic shift due to natural selection, since the younger age classes still have high frequencies. They conclude that the decrease in SS individuals is due to greater attrition of SS due to higher incidence and mortality. (This was actually clearer in the rebuttal letter than in the main text, because in the text the idea is spread out over several paragraphs with other concepts in between.) But the interesting yet glossed over part is that it isn't translating to increased *F frequencies in younger age classes. We can't tell why – if 225SS are dying after they reproduce, or if recruitment is lower for *F. But there needs to be more of a discussion of this. It is actually a really important finding and suggests that we might not expect evolution to come to the rescue here.	These comments – particularly the observation that our rebuttal conveyed these concepts more clearly than the manuscript text – motivated considerable reorganization of the text falling within lines 56–128 of the previous revision. The overall findings, data details, table, and figures have not changed and the writing itself was changed very little. Rather, we moved select sentences and rearranged several blocks of text to better group and present the genetic findings without interruption. The Results & Discussion storyline now unfolds more logically in parallel with the progression of our analyses of these field data, initially covering the apparent overall epidemic and demographic stability before probing the underlying patterns and their potential explanations. We did borrow a bit of text from the earlier rebuttal to help frame the discussion of attrition vs. genetic shift. In the course of making these changes we briefly address the dampened prospects for 'evolutionary rescue' suggested by the data, also noting our inability to fully explain why 225*F frequencies have not increased in younger age classes. Hopefully these small changes are sufficient to address the suggestions for additional treatment of this topic.
Additionally, a new study is referenced here now (ref 19), which is also comparing past and current trends at a site in Wyoming. There needs to be more of a discussion of how these 2 studies fit together. For example, they saw comparable changes in genotype, but didn't they also see an increase in prevalence? They didn't look at age structure. Based on your findings of higher attrition of SS individuals rather than a genetic shift, could this change the implications of their study?	Text edited to clarify that LaCava et al. (2021) reported on data from seven Wyoming mule deer herds (including but not limited to the South Converse herd). Their analyses did not specifically describe changes in prevalence and summary data limitations (see their Table 1) precluded our independent assessment of trends. However, the raw numerical differences in reported prevalence ranged from 2% to 30%, with the greatest change in prevalence occurring in a herd (Goshen Rim) showing a comparatively small change in 225*F genotype frequency (~+0.064). Although difficult to compare these studies directly, our findings may compel rethinking on the interpretation of so-called genetic shifts observed elsewhere. This has been noted in the newly revised draft.
225SS are implied to have lower survival because we aren't seeing as	This is a question we already had considered in the course of deciding how to assemble

many older individuals as before. If survival was lower in 2018-2020 compared to 2005-2008, they shouldn't be lumped together in Fig 3. They should be examined separately.	these data. Overall survival of infected deer did not differ between the two study periods (logrank test $P=0.087$), suggesting the disease course was unchanged but exposure & infection is now coming about earlier in life. We have modified the text slightly to clarify that the observed pattern is presumably more a function of earlier exposure resulting in an earlier age at disease-related death. In captivity, both earlier exposure and a higher exposure dose (which can result in a shorter disease course) likely work in concert to drive these patterns. Which may be why they're more dramatic in confinement settings. Text revised to clarify.
"The phenomenon of progressively earlier onset & mortality has been described in sustained epidemics of chronic wasting disease in captive mule deer and in scrapie-infected sheep flocks (references added to the revised paper)." In these previous studies was prevalence increasing or remaining the same?	Neither of the referenced studies reported apparent prevalence. The patterns of incidence (clinical) varied among the four scrapie-affected sheep flocks. Incidence did increase in the captive mule deer herd over time.
Line 152: This line is unclear. Is 0.16 the change in frequencies (A 16% increase?) Or did it change to 16% of these genotypes? If so where is the missing 14%? (Unidentifiable?)	We only mentioned the value "0.16" in line 72, so perhaps the line reference is a typo? However, we see how the source/interpretation of this value may be confusing as presented. This is the metric used by LaCava et al. (2021; see their Fig. 4) in illustrating the relationship between starting prevalence and observed "change" in 225*F genotype frequency in mule deer herds 11–18 years later. We have added detail in lines 72–73 to better explain that the 0.16 is the difference between the observed frequencies observed Table Mesa at two time points 13 years apart ($0.24-0.08=0.16$).